# Time-efficient physical activity intervention for older adolescents with disability: rationale and study protocol for the Burn 2 Learn adapted (B2La) cluster randomised controlled trial

Toby J Kable,[1,2,3] Angus A Leahy [ID],[1,2,3] Jordan J Smith,[1,2,3] Narelle Eather,[1,2,3] Nora Shields [ID],[4] Michael Noetel,[5] Chris Lonsdale,[6] Charles H Hillman,[7,8] Penny Reeves,[9,10] Christopher Oldmeadow [ID],[11] Sarah G Kennedy,[12] James Boyer,[13] Leisl Stimpson,[14] Pierre Comis,[14] Laura Roche,[1] David R Lubans [ID] [1,2,3,15]

For numbered affiliations see end of article.

**Correspondence to**
Professor David R Lubans;
David.Lubans@newcastle.edu.au

## ABSTRACT

**Introduction** Physical activity declines during adolescence, with the lowest levels of activity observed among those with disability. Schools are ideal settings to address this issue; however, few school-based interventions have been specifically designed for older adolescents with disability. Our aim is to investigate the effects of a school-based physical activity programme, involving high-intensity interval training (HIIT), on physical, mental and cognitive health in older adolescents with disability.

**Methods and analysis** We will evaluate the Burn 2 Learn adapted (B2La) intervention using a two-arm, parallel group, cluster randomised controlled trial with allocation occurring at the school level (treatment or waitlist control). Secondary schools will be recruited in two cohorts from New South Wales, Australia. We will aim to recruit 300 older adolescents (aged 15–19 years) with disability from 30 secondary schools (10 in cohort 1 and 20 in cohort 2). Schools allocated to the intervention group will deliver two HIIT sessions per week during scheduled specialist support classes. The sessions will include foundational aerobic and muscle strengthening exercises tailored to meet student needs. We will provide teachers with training, resources, and support to facilitate the delivery of the B2La programme. Study outcomes will be assessed at baseline, 6 months (primary endpoint), and 9 months. Our primary outcome is functional capacity assessed using the 6 min walk/push test. Secondary outcomes include physical activity, muscular fitness, body composition, cognitive function, quality of life, physical literacy, and on-task behaviour in the classroom. We will also conduct economic and process evaluations to determine cost-effectiveness, programme acceptability, implementation, adaptability, and sustainability in schools.

**Ethics and dissemination** This study has received approval from the University of Newcastle (H-2021–0262) and the New South Wales Department of Education (SERAP: 2021257) human research ethics committees. Findings will be published in peer-reviewed journals, and

## STRENGTHS AND LIMITATIONS OF THIS STUDY

⇒ Our cluster randomised controlled trial will be adequately powered to detect meaningful changes in the primary outcome functional capacity.
⇒ Informed by our pilot work, we have tailored the intervention and assessment processes to increase accessibility for the unique study population.
⇒ The Burn 2 Learn adapted intervention has been designed in consultation with adolescents with disability, and key stakeholders (ie, NSW Department of Education and Special Olympics Australia).
⇒ Having a unique study population with physical and/or intellectual limitations, not all participants will be able to complete all measures.
⇒ It might not be possible to blind assessors for all outcomes, as group allocation is often revealed by research participants and teachers during post-test assessments in school-based trials.

key stakeholders will be provided with a detailed report following the study.

**Trial registration number** Australian New Zealand Clinical Trials Registry Number: ACTRN12621000884808.

## INTRODUCTION

Disability is an umbrella term used to describe impairments (ie, problems in body function or structure), activity limitations (ie, difficulty in performing activities), and participation restrictions (ie, difficulty engaging in life situations).[1] Disability is a worldwide public health and human rights issue with 15% of the global population estimated to be living with disability.[2] These individuals often face widespread barriers to accessing health and related services, and have poorer health

outcomes than those without disability.[3] As noted in the WHO's Global Disability Action Plan, the burden of disability can be reduced by addressing the determinants of health, including participation in physical activity.[3] Individuals with disability are typically less physically active[4] and more likely to have co-occurring chronic and complex lifestyle diseases[5 6] than those without disability.

Young people with disability face many common and unique barriers to participation in physical activity. Previous research has identified a range of personal (ie, injury, lack of skills and time to exercise), social (ie, unsupportive peers and parents) and environmental (ie, inadequate accessibility and lack of appropriate programmes), barriers to participation in this population.[7 8] Conversely, factors shown to facilitate participation include having the time available, being involved in programmes that are adaptable, and exercising in a group with people of a similar age.[9] Parents also play a critical role in determining whether young people with disability are physically active.[8] Although parents typically acknowledge the value of physical activity and want their children to be active, many also express concerns about time commitments, balancing the needs of family members, and the suitability of programmes.[8] To date, the majority of physical activity interventions targeting young people with disability have been conducted in clinical, community, and home settings.[10]

Schools are ideal for physical activity promotion, as they provide access to the adolescent population and have the necessary equipment, facilities, and personnel to deliver programmes.[11] Physical education (PE) is the primary means of physical activity promotion in schools, and there is a large body of research focusing on the inclusion of children and adolescents with disability in PE classes.[12] Although teachers typically advocate for inclusion in PE, many lack the confidence and competence to successfully involve students with disability in ways that truly benefit their physical literacy.[12] Moreover, students with disability often feel marginalised when participating in mainstream PE classes, commonly reporting feelings of social isolation, bullying, and negative social comparisons.[13] Regardless, simply integrating students with disability into mainstream PE is not enough to produce meaningful changes in health, particularly in the final years of school where there is no mandatory physical activity for students.[14] In light of this, there is a need for innovative school-based physical activity interventions designed specifically for older adolescents with disability.

Implementing health promotion interventions with older adolescents is challenging, and lack of time is a major barrier to physical activity promotion for this age group.[15] For older adolescents with disability, the final years of school also involves participation in programmes facilitating transition into postschool pathways (eg, community access and transition to work programmes). Given that adolescents with disability have many competing needs as they prepare for life after schooling, school-based physical activity programmes will have the best chances of adoption if they do not require a substantial time commitment. However, physical activity programmes that provide only a small 'dose' of activity are also unlikely to have meaningful health benefits, that is, unless the physical activity offered is of 'high intensity'.

High-intensity interval training (HIIT) is a time-efficient strategy for improving physical, mental and cognitive health in typically developing adolescents.[16 17] HIIT sessions generally consist of several short bouts of vigorous activity interspersed with brief periods of light activity or rest. HIIT allows participants to experience similar benefits to other modes of exercise, in less time. Previous studies evaluating school-based HIIT programmes for adolescents with disability have shown improvements in physical health (ie, body composition, aerobic fitness) but have been delivered by physiotherapists or experienced physical educators.[18 19] Such programmes have limited scalability due to ongoing costs required for intervention implementation. To enhance programme scalability, having classroom teachers implement the intervention has greater potential to change school practice.

We recently conducted a large-scale evaluation of the first teacher-facilitated school-based HIIT intervention, known as Burn 2 Learn (B2L), for older adolescents in mainstream schools.[20–23] Briefly, teachers were trained to deliver two to three HIIT sessions per week for 16 weeks during students' regular academic lessons. Positive effects were observed for the primary outcome of cardiorespiratory fitness, as well as a range of secondary outcomes (eg, muscular fitness, mental health, and classroom engagement). Following the success of the intervention, our research team was approached by a local school to adapt the B2L intervention for students with disability. We subsequently conducted a pilot study of the Burn 2 Learn adapted (B2La) programme in one secondary school.[24] We found it was feasible to train special and inclusive education teachers to deliver the B2La sessions, which were well received by teachers and students. We also found preliminary support for programme efficacy for improving functional capacity and muscular fitness. Following our successful feasibility study, we partnered with the NSW Department of Education and Special Olympics Australia to refine and evaluate B2La in a larger effectiveness trial.

## Study objectives

The primary aim of this trial is to determine the effect of the B2La intervention on functional capacity (primary outcome) in older adolescents with disability. Secondary outcomes include physical activity (accelerometers), muscular fitness, body composition, cognitive function, mental health, physical literacy and on-task classroom behaviour. We will also conduct economic and process evaluations to determine cost-effectiveness, programme efficiency, acceptability, implementation, adaptability, and practicality.

## METHODS

### Study design

Our trial is registered with the Australian New Zealand Clinical Trials Registry (ACTRN12621000884808). The reporting of this trial will adhere to the CONSORT[25] guidelines. The B2La intervention will be evaluated using a two-arm parallel group cluster randomised controlled trial (RCT) with a treatment group and wait-list control group. Assessments will occur at baseline, 6 months (primary endpoint) and 9 months from baseline (secondary endpoint). The RCT will be conducted with two cohorts, one starting in 2022 (10 schools; five intervention and five control) and another starting in 2023 (20 schools; 10 intervention and 10 control). Baseline data collection will occur in the school term preceding intervention delivery (ie, term 1 (February–April 2022 and 2023). The intervention will be delivered during terms 2, 3 and 4 (April–November 2022 and 2023). Immediate postintervention data collection (ie, ~6 months) will occur at the end of term 3 (August–September 2022 and 2023), and follow-up assessments (ie, ~9 months) will be completed in term 4 (November–December 2022 and 2023).

### School recruitment and selection

NSW Government, Catholic and independent secondary schools with student cohorts that include older adolescents (ie, grades 10–12, students aged 15–19 years) with disability will be eligible to participate. Schools will include both mainstream schools with specialist support classes and schools for specific purposes (SSPs). SSPs are dedicated schools for students with moderate-to-high learning and support needs. Eligible schools will be identified, and an expression of interest directed to the school principal. Interested schools will then liaise with the project manager to address any questions or concerns they have prior to returning informed consent.

### Participants

Students at the study schools will be eligible to participate if they are: (1) in Grades 10 to 12 (15–19 years) and identify as living with disability (including neurodevelopmental disability, physical, intellectual or sensory disabilities), (2) able to follow simple verbal instructions in English (as determined by the Index of Social Competence)[26] and (3) able to participate in vigorous intensity exercise (wheelchair users will be eligible). We will aim to recruit 10 students from each school. We will also recruit two special and inclusive education teachers per school, who will act as school champions and facilitate the delivery of B2La sessions. Special and inclusive education teachers develop and deliver specialised learning programmes for students with a range of disabilities and learning difficulties.

### Sample size and power calculation

Power calculations were based on the primary outcome of functional capacity, assessed using the 6 min walk test (6MWT), which has good reliability in adolescents with disability (intraclass corelation coefficient (ICC)=0.82).[27] A 6 min push test will be administered for wheelchair users. Although adolescent data are lacking, studies conducted among adult populations with chronic health conditions have reported minimal clinically important differences (MCID) of 24–44 m using the 6MWT.[28] In our pilot study, we observed a large increase in distance covered from baseline to immediately after the intervention period (163±131 m). However, our pilot study did not include a control group, and effects are typically smaller in effectiveness trials compared with pilot studies.[29] Based on the pilot data, we estimate a treatment effect of 80 m will represent a MCID in our population. Through simulations (n=10 000) and using data from our pilot study (ie, baseline post-test correlation of $r$=0.60, SD of 90 m and an ICC of 0.2), we have determined we will require a sample of 30 schools with seven participants per school. This sample size will be enough to detect an MCID of 80 m with 90% power at a 5% significance level. Allowing for 30% loss to follow-up at 6 months, we will aim to recruit 10 students from 30 schools (total sample size of 300).

### Blinding and randomisation

Randomisation will occur within each cohort once consenting schools have completed baseline assessments. Schools will be matched as closely as possible based on the following characteristics in this order: (1) school type (ie, mainstream school support class/SSP), (2) school sector (ie, government/Catholic/independent), (3) geographic location (ie, region, rural/urban, coastal/inland) and (4) student population educational advantage (ie, using the Index of Community Socio-Educational Advantage).[30] Matched schools will be randomised to the intervention or waitlist control group using a random number producing algorithm by an independent statistical analysis service – Clinical Research Design, Information Technology and Statistical Support run by the Hunter Medical Research Institute. One school from each pair will be allocated to the B2La condition and the other to the waitlist control condition. Schools randomised to the intervention group will deliver the B2La programme during the study period, whereas schools allocated to the waitlist control group will continue with usual school practice (ie, normal curricular lessons) for the duration of the study period and will receive the intervention the following year. We decided to use a waitlist control design, rather than an attention-matched placebo because: (1) our research team will have minimal contact with students, and (2) our findings will have greater external validity, as participants in the control group will receive 'usual practice'.

### Patient and public involvement

Following the B2L cluster RCT, our research team was asked to adapt the intervention for students with disability. We subsequently conducted a feasibility study in one secondary school in Newcastle (n=16 students).[24] Participating students and teachers were invited to provide

feedback on the intervention and suggestions for further improvement. This feedback was then used to refine intervention components (eg, exercise sessions) and develop implementation strategies (eg, professional learning for teachers). We then partnered with the NSW Department of Education and Special Olympics Australia to create B2La. We conducted further testing with teachers and students with disability before progressing to this trial.

## Intervention

### Intervention delivery

The B2La intervention will be delivered in four phases:
1. Laying the foundation (weeks 1–4).
2. Developing a routine (weeks 5–9).
3. Maintaining student interest (weeks 10–16).
4. Moving towards independence (week 17 onwards).

In phases 1–3 (16 weeks), teachers will facilitate the delivery of at least two HIIT sessions/week during lesson time. Phase 1 will start with a 4 week block to familiarise students with the B2La session structure and support

resources and to develop the foundational exercise skills that are used within the HIIT sessions utilising the B2La technique cards (figure 1). During this phase, students will participate in two HIIT workouts: indoor HIIT and power HIIT (as shown in figure 2). These basic HIIT workouts do not require additional sport equipment or partner interaction. This will allow students to develop their movement skill competency.

In phases 2 and 3 (weeks 5–16), the number of foundational exercises used within the HIIT sessions will increase as students become more confident with the exercises and session routine. During this phase, there will be an increase in the work interval within the HIIT sessions. Students will also be introduced to novel HIIT-themed workouts, including: soccer HIIT and basketball HIIT (phase 2) and judo HIIT, cricket HIIT and custom HIIT (phase 3). During phases 2 and 3, teachers will be encouraged to increase the amount of autonomy provided to students (eg, choice of B2La session). In phase 4 (week

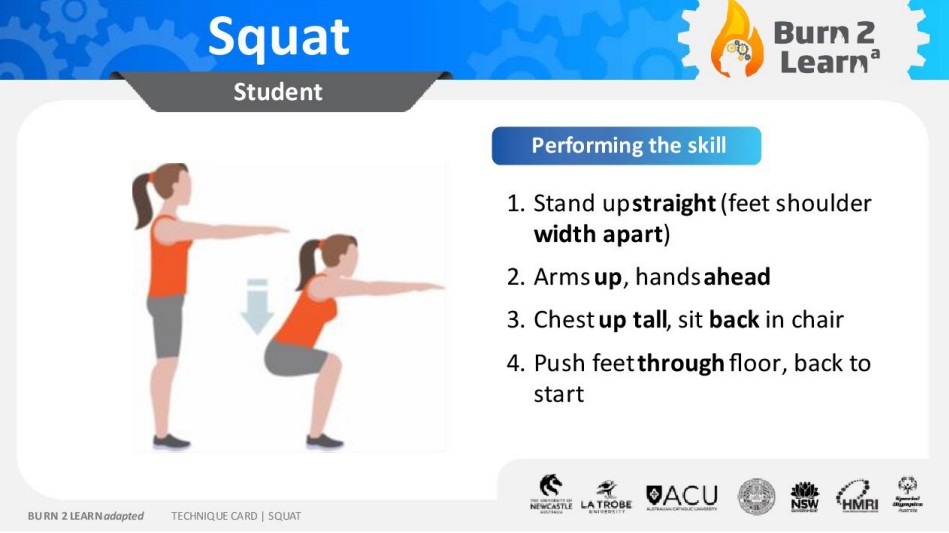

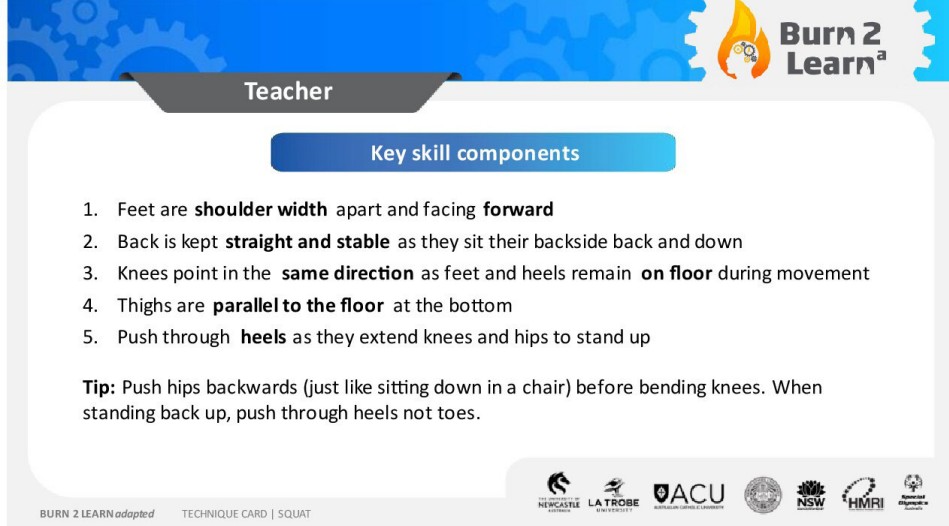

**Figure 1** Example of B2La HIIT technique card. This figure was created by the lead investigator.

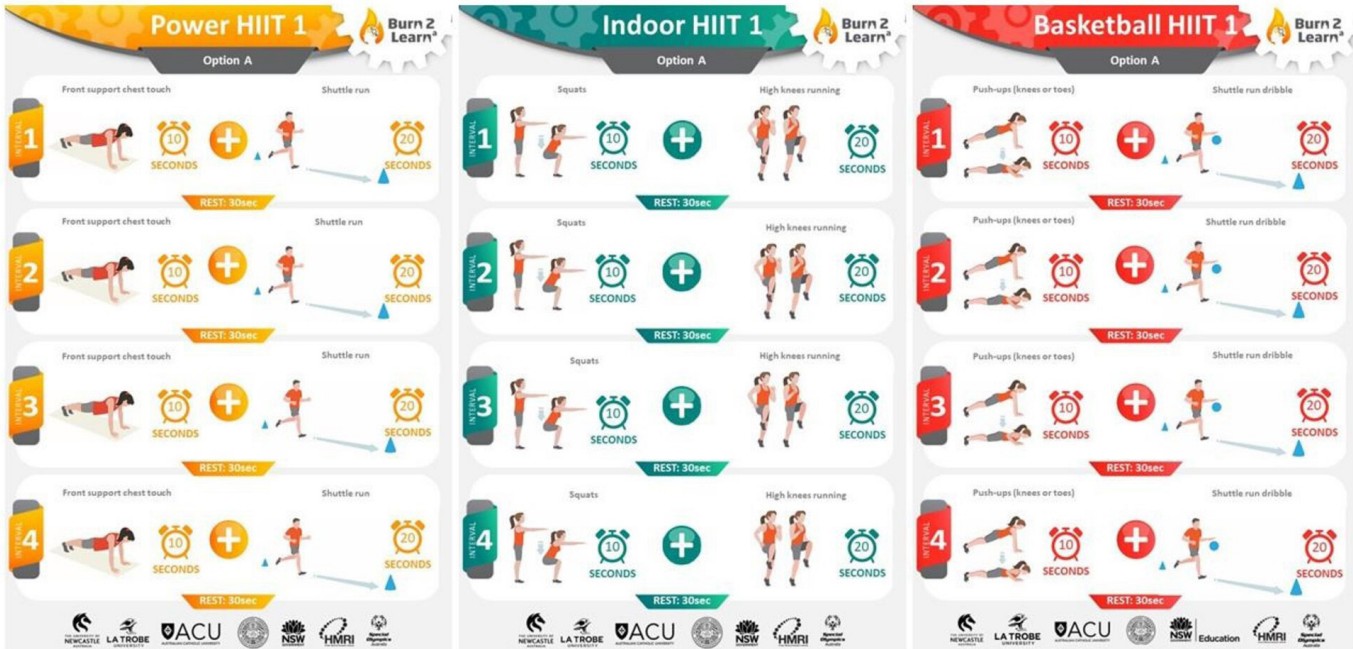

**Figure 2** Examples of B2La HIITsession cards.This figure was created by the lead investigator.

17 onwards), students will be encouraged to engage in physical activity sessions of interest outside of school (eg, at home, the park), but teachers may continue to facilitate B2La sessions during lessons if they choose. Students will have their own B2La goal setting activity booklet.

### Intervention components
The B2La intervention includes the following components: student information seminar, school-based HIIT sessions, smartphone application (app), goal setting activity booklet and parental support videos.

### *Student information seminar*
This seminar will be delivered by teachers and provide students with an overview of B2La. The seminar will focus on the barriers and benefits of physical activity for adolescents with disability[4], as well as evidence-based behaviour change techniques (eg, self-monitoring, self-assessment, and goal setting). Teachers will be provided with a Power-Point presentation template with embedded videos developed specifically for this project.

### *School-based HIIT sessions*
Sessions will be delivered during scheduled 'Learning Support Lessons', a time period when adolescents with disability are working separately to those without disability (for those attending mainstream schools). Students with special needs in Australian mainstream secondary schools typically attend three, 2-hour support lessons per week. Teachers will be asked to facilitate the delivery of at least two exercise sessions per week in phases 1–3. Adapted from the original B2L intervention, teachers/participants will be able to select from predesigned themed HIIT workouts that include a combination of foundational

resistance exercises (ie, push up, squat, front support, and lunge) and aerobic exercises (ie, shuttle run and high knees running on the spot). The sessions last 10–20 min including an appropriate warm-up. Due to the wide range of abilities expected from the participants, teachers will be encouraged to adapt the specific types of exercises and task complexity for each student. The task complexity and variations within the HIIT sessions will progressively increase over the study period. To monitor exercise intensity, students will be equipped with heart rate sensors (Polar Verity Sense) that will pair with a purpose-built iPad application (hereafter 'app') using Bluetooth connectivity. Students will be encouraged to reach a target intensity of ≥80% of age-predicted heart rate maximum during the work intervals. As demonstrated in our feasibility study, this heart rate target is achievable for students with disability.[24]

### *Smartphone app*
Teachers and students will be provided with access to a bespoke smartphone app available via both Android and iOS operating systems. The app includes: (1) a teacher version that allows whole-class heart rate monitoring during 'class' sessions, (2) descriptions and depictions of exercise sessions, (3) options for 'solo' or 'group' sessions, (4) timer, audible prompts and display of heart rate using Bluetooth-synced heart rate monitors during HIIT sessions, and (5) personalised reports outlining heart rate. For students who do not own a smartphone, access to the B2L app will be provided to parents. During the professional development workshop, teachers will be encouraged to deliver school-based sessions using the teacher version that allows whole-class heart rate

monitoring. Students will be encouraged to use the app during activity sessions outside of the school setting.

### Goal setting activity booklet
Students will be provided with a goal setting activity booklet and encouraged to set physical activity and fitness goals over the study period.

### Parent support videos
Parents will receive two e-newsletters containing links to video overviews of B2La, the benefits and barriers of physical activity for individuals with disability and strategies to support their children's participation in physical activity outside of school. The e-newsletters will be delivered to parents in phases 1 and 2 of the intervention.

### Intervention conceptual model and theoretical frameworks
B2La was guided by the conceptual model proposed by Lubans and colleagues,[31] which includes four complementary tenets that are fundamental to the successful scale-up of adolescent HIIT interventions.

### Opportunity
The B2La sessions will be delivered during scheduled 'Specialist Support Classes' (students typically attend 3×2-hour support lessons/week). These classes cater for students with moderate-to-high learning and support needs, including students with intellectual disability, mental health issues, autism, physical disability, sensory impairment and behaviour disorders. Based on our formative research with teachers and the NSW Department of Education, 'Specialist Support Classes' represent an ideal 'new' opportunity[32] for the delivery of B2La.

### Design
The school-based exercise sessions will provide participants with opportunities to collaboratively develop their exercise competence, confidence, and knowledge. Participants will also be provided with opportunities to design and run their own HIIT sessions. The study information seminar will focus on the benefits and barriers of physical activity for individuals with disability[4] as well as evidence-based behaviour change techniques.

### Delivery
B2La has been guided by Self-Determination Theory (SDT) to enhance students' autonomous motivation for physical activity.[33] Aligned with SDT, teachers will be provided with training and support to deliver the B2La sessions using the 'Supportive, Active, Autonomous, Fair and Enjoyable' (SAAFE) principles.[34] Participants' need for autonomy will be satisfied by providing opportunities for choice within sessions (eg, type of activity, music playing and training partner). Competence will be satisfied using positive and skill-specific feedback from teachers, with a focus on effort over performance (via heart rate feedback). Schools will also be provided with resources designed to support the accessibility, engagement, and development of exercise skills (eg, technique

cards, B2L app). Teachers will use a variety of strategies to enhance group cohesion and satisfy students' needs for relatedness during HIIT sessions (ie, encouraging supportive behaviour among students such as 'high fives' and facilitating partner work).[35]

### Support
The implementation of B2La will be supported by the Consolidated Framework for Implementation Research (CFIR).[36] Strategies used to facilitate the implementation of the B2La intervention will cover the five CFIR domains: intervention characteristics, outer setting (educational authorities), inner setting (schools), characteristics of individuals (teachers) and the implementation process (see table 1).

Teachers recruited as school champions will attend a full day professional learning workshop led by members of the research team. The workshop will provide teachers with the training and resources needed to facilitate school-based HIIT sessions. The workshop will involve a combination of theoretical (ie, programme rationale, benefits of HIIT and school implementation plan) and practical (eg, participation in a B2La HIIT session, peer assessment of exercise technique and overview of how to use programme resources) activities.

### Measures and data collection
Trained research assistants, blinded to group allocation at all time-points, will conduct assessments for the primary outcome. Questionnaires will be completed with the assistance of research assistants using electronic tablets. Physical assessments will be conducted in a sensitive manner by a research assistant of the same sex where possible. Standard demographic information (eg, age, sex, ethnicity, country of birth, residential postcode and parent/caregivers' education level) will be collected at baseline. All measurements will be conducted at baseline, 6 months postbaseline (primary endpoint) and 9 months postbaseline. The only exception will be students' on-task behaviour, which will be assessed at baseline and midintervention (3 months postbaseline), and cognitive function, which will be assessed at baseline and 6 months only. Of note, due to physical and intellectual limitations, not all participants will be able to complete all measures, and modifications will be made, as necessary.

### Primary outcome
#### Functional capacity
Consistent with previous physical activity interventions targeting youth with disability,[37] our primary outcome is functional capacity, assessed using the 6MWT,[38] which has good reliability in adolescents with disability (ICC=0.82).[27] The 6 min push test will be used for students who self-propel a wheelchair.[39] Students will be instructed to cover as much distance as possible in 6 min, and the distance (in metres) covered will be documented.

**Table 1** Strategies used to facilitate implementation in the Burn 2 Learn (B2L) adapted (B2La) intervention

| Domains | Constructs | Strategies |
|---|---|---|
| B2La intervention characteristics | Evidence strength and quality | Findings from B2L cluster RCT and B2La feasibility study used in promotional and training materials. |
| | Adaptability | Flexible intervention delivery model (ie, during class-time or breaks, or between classes) requiring minimal access to facilities (ie, can be done in the classroom) and equipment (ie, body weight exercises). |
| | Complexity | Time-efficient intervention requiring only two or three 15–20 min sessions per week. |
| | Design quality and packaging | B2La programme resources developed by a professional graphic designer. Multimedia designed using evidence-based principles for learning. |
| Outer setting (educational authorities) | Partnerships and investment | Partnership with the NSW Department of Education and Special Olympics Australia. |
| | External policy and incentives | Professional learning accreditation with state-based educational standards authority. |
| Inner setting (schools) | School culture | Teachers will be encouraged to give a presentation to school staff focused on the benefits of activity for students' mental health and academic outcomes. |
| | Leadership engagement | Teachers and external change agents will meet with the school principal to ensure commitment. |
| | Equipment | Schools will be provided with an equipment pack (~$A2000). |
| | Relative priority | Promoted to schools as strategy to improve students' cognitive function and mental health. |
| Characteristics of individuals (teachers) | Self-efficacy, knowledge and beliefs (teacher) | Full day professional development workshop provided for teachers. Online version of workshop available. |
| | Perceived barriers (students) | Designed to be time-efficient, and motivating for students, through the SAAFE teaching principles. |
| Implementaton process | Planning for implementation | Teachers required to complete an action plan to support B2La implementation in their school. |
| | Champions | Recruitment of two school champions at each intervention school. |
| | External support agents | Schools will be allocated external change agent, who will visit twice for planning and evaluation. |
| | Evaluation and feedback | External change agents will conduct session observations and provide feedback to teachers. |

SAAFE, Supportive, Active, Autonomous, Fair and Enjoyable.

## Secondary outcomes

### Physical activity

Participants will be instructed to wear an ActiGraph GT9X Link accelerometer on their non-dominant wrist for 24 hours/day (even when bathing, swimming and sleeping) for a period of seven consecutive days (3 days' minimum wear time). School hour, weekday and weekend (ie, mean minutes per day) physical activity will be calculated separately, using existing thresholds for categorising physical activity intensity.[22]

### Muscular fitness

Lower body muscular endurance will be assessed using the 30 s sit-to-stand test.[40] From a seated position, students will attempt to stand up and sit back down on a 44 cm high bench seat as many times as possible in 30 s.[18] A modified version of the 90° push-up test will be used to assess upper body muscular endurance.[41] All students will be instructed to perform as many push-ups as possible on their knees.

### Body composition

Body weight and height will be measured using a portable digital scale (A&D Medical UC-352-BLE Digital Scales) and a portable stadiometer (Seca 213 Portable Height Measuring Rod Stadiometer), respectively. Body mass index (BMI) will be calculated using the standard formula (weight[kg]/height[m]$^2$). Age-specific and sex-specific BMI z-scores will be calculated, and participants will be classified into weight categories according to International Obesity Task Force cut-offs.[42]

### Cognitive function

This will be assessed with electronic tablets using the cognitive portion of the National Institutes of Health (NIH) Toolbox.[43] The Toolbox has been used with children and adults with Fragile X syndrome, Down syndrome and intellectual disabilities, with tests demonstrating good to excellent reliability and feasibility.[43 44] Participants will complete the Flanker (inhibition), list sorting (working memory), and dimensional change card sort (cognitive flexibility) tasks.

### Quality of life

Health-related quality of life will be assessed using the Child Health Utility 9-Dimensions,[45] which includes nine items (worried, sad, pain, tired, annoyed, schoolwork or work, sleep, daily routine and activities), and each item is scored on a five-point scale.

## Physical literacy

Autonomous motivation for physical activity will be assessed using identified and intrinsic subscales from the 'Behavioural Regulations in Exercise Questionnaire-2'.[46] Confidence will be assessed using the validated six-item High-Intensity Interval Training Self-efficacy Questionnaire.[47] Competence will be assessed using video analysis of a selection of skills from the Resistance Training Skills Battery (ie, push-ups, lunge, squat and front support chest touch), which has been validated among typically developing adolescents[48] and among children with varying degrees of motor skill proficiency.[49]

## Externalising behaviours

Teachers will complete a Student Behaviour Questionnaire[50] for each student at baseline, 6 months, and 9 months. The questionnaire consists of 10 statements, regarding students' classroom behaviours, observed over the previous 6 months, which are rated using a 3-point Likert scale. The items have been adapted from the strengths and difficulties questionnaire externalising subscale.

## On-task behaviour

To determine the acute effect of the B2La intervention on students' behaviour in the classroom, observations will be conducted by trained research assistants at baseline and midintervention (3 months) using established methods.[51] The assessment includes a 30 min observation period where research assistants will assess the on-task and off-task behaviour of six randomly selected students (5 min per student). Observation and recording are completed in 15 s intervals (20 observations per student), and teachers and students will not know who is being observed during the assessment.

## Economic evaluation

We will assess the efficiency and affordability of the intervention using cost-effectiveness/cost utility analysis and budget impact, respectively, conducted from a public finance perspective. The effectiveness measure will be based on the primary outcome (6MWT). Transformation of the Child Health Utility 9-Dimensions data will be employed in a cost utility analysis. The resource use and costs for the intervention and usual practice will be prospectively measured and derived from project records (staff and consumables), teacher surveys and school records. Additional costs in the intervention group are anticipated to be labour (implementation support), programme development and training costs. The cost-effectiveness analysis will be conducted on a 'within trial' basis, that is, over the 6-month study period, comparing incremental costs and outcomes. Affordability of the programme will be calculated using budget impact analysis, over a standard accounting cycle and is designed to assist decision making in schools and hence assist the translation of cost-effective and affordable programmes. Scenario analysis will assess the costs to implement the programme at scale across NSW. Reporting for the economic analysis will adhere to the Consolidated Health Economic Evaluation Reporting Standards (CHEERS) statement.[52]

## Process evaluation

We will conduct a process evaluation to determine programme acceptability, implementation, adaptability, and sustainability in schools.

### Acceptability

We will conduct focus groups to determine teachers' and participants' (ie, students) perceptions of, and experiences with, the intervention. Teachers will also complete the Acceptability of Intervention Measure, Intervention Appropriateness Measure, and Feasibility of Intervention Measure.[53]

### Implementation

Teachers will be asked to record their delivery of B2La sessions using the teacher handbook. We will also track the number of sessions delivered using the B2La smartphone app. Members of our research team will conduct two session observations (using the SAAFE checklist) at each school to determine intervention fidelity. Finally, participants' mean heart rate during sessions will be collected using the B2L app.

### Adaptability

Teachers will be asked to reflect on how they adapted the intervention in the focus groups. This will include adaptions in relation to the characteristics of the school, class, and students.

### Sustainability

Sustainability will be explored in the focus groups and via teacher and participant postprogramme evaluation questionnaires. Teachers will report their intention to deliver B2L in the future and complete an adapted version of the Program Sustainability Assessment Tool.[54 55] Students will report their intention participate in HIIT in the 2 months following programme completion.

## Statistical analyses

Blinded analyses of the primary and secondary outcomes will be conducted by an independent statistician, using linear mixed models SAS V.9.1 (SAS Institute Inc), with alpha levels set at $p < 0.05$. The models will be used to assess the impact of group (B2La or control), time (treated as categorical with levels baseline, 6 months and 9 months) and the group-by-time interaction. The models will include a random intercept for participant to account for the repeated measures for each participant and a random intercept for school to account for the clustered design. The primary endpoint of the study will be 6 months from baseline. Least square mean differences between the treatment groups will be presented at both follow-up time points, with 95% CIs and p values. Compared with complete case analyses, mixed models include available data for all participants and are thus both more efficient and robust to bias. Mixed model analyses are consistent with the intention-to-treat principle, assuming the data are missing at random. The validity of this assumption will be explored by assessing relationships between missingness

and observed values. We will conduct two sensitivity analyses for the primary outcome: (1) multiple imputation (assuming data are missing at random) and (2) complete case analysis (assuming data are missing completely at random). Four potential moderators (ie, socio-economic status, sex, initial weight status and disability type) will be explored using interaction terms (ie, time-by-treatment-by-moderator). If an interaction term is significant (p<0.1), sub group analyses will be conducted.

## Data monitoring

An internal monitoring committee consisting of DRL (lead), AAL, and the project manager will oversee the conduct of the study and manage any data or safety issues that may arise. All entered data will be deidentified using participant codes and will be stored electronically in a password-protected drive at the University of Newcastle. Data will be checked for implausible values, and 20% of the data will be entered twice to confirm accuracy. It is not expected that participants will be at any greater risk of adverse events than they would be when participating in other types of school-based physical activity. However, the teacher handbook includes a section for teachers to report any injuries or adverse events that may occur. Any adverse events will be documented and reported to the relevant ethics committee. Any amendments to the study protocols will be publicly available via the Australian and New Zealand Clinical Trials Registry (Trial number: ACTRN12621000884808). We have not included any formal guidelines for stopping the trial early, as we have not planned a formal interim analysis of the primary outcome.

## DISCUSSION

B2La has been designed to provide older adolescents with disability an opportunity to be active at school but also focuses on developing their physical literacy (ie, physical competence, confidence, knowledge, and motivation) to engage in vigorous physical activity. Importantly, our research team will provide teachers with training and support to ensure that the programme is delivered in an engaging manner that supports students' autonomous motivation to be active across the lifespan. Most HIIT studies have been delivered by researchers in controlled settings to establish efficacy, with little consideration of how they will work in the 'real world'.[31] By comparison, B2La was designed with scale-up in mind using the Consolidated Framework for Implementation Research to support implementation and sustainability. This may help to reduce the 'voltage drop' that typically occurs as interventions progress from efficacy to effectiveness to dissemination.[29 56–58]

The strengths of our study include the cluster RCT design. We also consider our intervention design to be a strength, as it was developed in consultation with adolescents with disability and key stakeholders (ie, NSW Department of Education and Special Olympics Australia). Our comprehensive assessment of physical, mental, and cognitive health is an additional study strength. However, there are potential limitations that should be noted. First, COVID-19 is still a major problem in Australian schools, resulting in high levels of teacher and student absenteeism. This is likely to affect recruitment, data collection and intervention implementation. Second, having a unique study population with physical and/or intellectual limitations, not all participants will be able to complete all measures. Finally, it might not be possible to retain the blinding of all research assistants at post-test assessments.

## ETHICS AND DISSEMINATION

Ethics approval for this cluster RCT was obtained from the Human Research Ethics Committee of the University of Newcastle, Australia (H-2021–0262) and the NSW Department of Education and Communities (SERAP:2021257). School principals, teachers, parents and students will all provide informed written consent prior to enrolment. Example participant information and consent forms are provided in our online supplemental materials. The full protocol, participant-level dataset and statistical code will be available on request from DRL. We will publish our findings in peer-reviewed journals and provide the NSW Department of Education and all participating schools with a detailed report at the conclusion of the trial. If the intervention is successful, we will support dissemination via a series of professional learning workshops for teachers in NSW schools.

**Author affiliations**
[1]School of Education, University of Newcastle, Callaghan, New South Wales, Australia
[2]Active Living Program, Hunter Medical Research Institute, New Lambton, New South Wales, Australia
[3]Centre for Active Living, University of Newcastle, Callaghan, New South Wales, Australia
[4]La Trobe University, Melbourne, Victoria, Australia
[5]School of Behavioural and Health Sciences, Australian Catholic University - Brisbane Campus, Banyo, Queensland, Australia
[6]Institute for Positive Psychology and Education, Australian Catholic University - North Sydney Campus, North Sydney, New South Wales, Australia
[7]Department of Psychology, Northeastern University, Boston, Massachusetts, USA
[8]Department of Physical Therapy, Movement and Rehabilitation Sciences, Northeastern University, Boston, Massachusetts, USA
[9]Health Research Economics, Hunter Medical Research Institute, New Lambton, New South Wales, Australia
[10]School of Medicine and Public Health, University of Newcastle, Callaghan, New South Wales, Australia
[11]Clinical Research Design and Statistics, Hunter Medical Research Institute, New Lambton, New South Wales, Australia
[12]School of Health Sciences, Western Sydney University, Kingswood, New South Wales, Australia
[13]School Sport Unit, NSW Department of Education, Sydney, New South Wales, Australia
[14]Special Olympics Australia, Sydney, New South Wales, Australia
[15]Faculty of Sport and Health Sciences, University of Jyväskylä, Jyväskylä, New South Wales, Finland

**Contributors** TJK: investigation, resources, data curation and writing – original draft. AAL: methodology, investigation, resources, data curation, writing – review and editing and funding acquisition. JJS and NE: methodology, investigation, resources, writing – review and editing and funding acquisition. NS and MN: methodology, resources, writing – review and editing and funding acquisition. CL and CH: methodology, writing – review and editing and funding acquisition. PR: writing – review and editing, guided statistical analysis and funding acquisition. CO: writing – review and editing and guided statistical analysis. SK, JB, LS and PC: resources, writing – review and editing. LR: resources, data curation, writing – review and editing. DRL: conceptualisation, methodology, investigation, resources, writing – original draft, supervision and funding acquisition.

**Funding** The study is funded by Medical Research Future Fund (APP2007095). DRL is funded by a National Health and Medical Research Council Senior Research Fellowship (APP1154507). This project is supported and co-designed with the NSW Department of Education and Special Olympics Australia.

**Disclaimer** The study funders will have no role in data collection, analysis, interpretation or writing. Nor will they influence over the publication of findings.

**Competing interests** None declared.

**Patient and public involvement** Patients and/or the public were involved in the design, or conduct, or reporting, or dissemination plans of this research. Refer to the Methods section for further details.

**Patient consent for publication** Not applicable.

**Provenance and peer review** Not commissioned; externally peer reviewed.

**ORCID iDs**
Angus A Leahy http://orcid.org/0000-0003-2147-9420
Nora Shields http://orcid.org/0000-0002-6840-2378
Christopher Oldmeadow http://orcid.org/0000-0001-6104-1322
David R Lubans http://orcid.org/0000-0002-0204-8257

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
