## [Reviewer comments · BMJ Open]

ARTICLE DETAILS

TITLE (PROVISIONAL)	Time-efficient physical activity intervention for older adolescents with disability: Rationale and study protocol for the Burn 2 Learn adapted (B2La) cluster randomised controlled trial
AUTHORS	Kable, Toby; Leahy, Angus; Smith, Jordan; Eather, Narelle; Shields, Nora; Noetel, Michael; Lonsdale, Chris; Hillman, Charles; Reeves, Penny; Oldmeadow, Christopher; Kennedy, Sarah; Boyer, James; Stimpson, Leisl; Comis, Pierre; Roche, Laura; Lubans, David

VERSION 1 – REVIEW

REVIEWER	Yoshiharu Fukuda Yamaguchi University, Community Health and Medicine
REVIEW RETURNED	30-Jun-2022

GENERAL COMMENTS	The study is very interesting, and the manuscript was written in term of background and methods. I look forward to the results.
---

REVIEWER	Francisco J Tarazona Hospital Universitario de la Ribera, Geriatric Medicine
REVIEW RETURNED	01-Jul-2022

GENERAL COMMENTS	The authors have presented an interesting protocol of a randomized clinical trial entitled "Time-efficient physical activity intervention for older adolescents with disability: Rationale and study protocol for the Burn 2 Learn adapted (B2La) cluster randomized controlled trial". The structure of the information provided in the abstract and material and methods sections are adequate, written fluently and precisely. However, the reviewer advises reducing the size of the introduction, properly detailing the limitations section in the discussion section and also explain the results that the authors expect to find.
---

VERSION 1 – AUTHOR RESPONSE

R1.1: The study is very interesting, and the manuscript was written in term of background and methods. I look forward to the results.

Thank you for the positive feedback.

R2.1 The authors have presented an interesting protocol of a randomized clinical trial entitled "Time-efficient physical activity intervention for older adolescents with disability: Rationale and study protocol for the Burn 2 Learn adapted (B2La) cluster randomized controlled trial". The structure of the information provided in the abstract and material and methods sections are adequate, written fluently and precisely. However, the reviewer advises reducing the size

of the introduction, properly detailing the limitations section in the discussion section and also explain the results that the authors expect to find.

We thank the Reviewer for taking the time to review our manuscript. Based on feedback from the Editor, we have decided to leave the introduction as is, and not include our expected study findings. We now discuss the strengths and limitations in the discussion section (see response to comment E.5b).